# A Bag of Tricks for Few-Shot Class-Incremental Learning

**Shuvendu Roy**[1,2]*, **Chunjong Park**[1], **Aldi Fahrezi**[1], **Ali Etemad**[1,2]†
[1]**Google Research**
[2]**Queen's University, Canada**
shuvendu.roy@queensu.ca, pcj@google.com, aldifahrezi@google.com, ali.etemad@queensu.ca

**Reviewed on OpenReview:** `https://openreview.net/forum?id=DiyYf1Kcdt`

## Abstract

We present a bag of tricks framework for few-shot class-incremental learning (FSCIL), which is a challenging form of continual learning that involves continuous adaptation to new tasks with limited samples. FSCIL requires both stability and adaptability, i.e., preserving proficiency in previously learned tasks while learning new ones. Our proposed bag of tricks brings together six key and highly influential techniques that improve stability, adaptability, and overall performance under a unified framework for FSCIL. We organize these tricks into three categories: stability tricks, adaptability tricks, and training tricks. Stability tricks aim to mitigate the forgetting of previously learned classes by enhancing the separation between the embeddings of learned classes and minimizing interference when learning new ones. On the other hand, adaptability tricks focus on the effective learning of new classes. Finally, training tricks improve the overall performance without compromising stability or adaptability. We perform extensive experiments on three benchmark datasets, CIFAR-100, CUB-200, and miniIMageNet, to evaluate the impact of our proposed framework. Our detailed analysis shows that our approach substantially improves both stability and adaptability, establishing a new state-of-the-art by outperforming prior works in the area. We believe our method provides a go-to solution and establishes a robust baseline for future research in this area.

## 1 Introduction

Continual learning is a machine learning paradigm that focuses on the ability of a model to learn new knowledge, without forgetting what it previously learned. In real-world applications, machine learning models often encounter scenarios where they must adapt to novel classes with only a limited number of samples available for learning. This scenario has inspired the introduction of an exciting paradigm called Few-Shot Class Incremental Learning (FSCIL) (Tao et al., 2020). Existing literature (Tao et al., 2020; Zhou et al., 2022a) has demonstrated that traditional continual learning approaches are ineffective in FSCIL, primarily due to the scarcity of labelled data during incremental learning sessions. This data limitation often leads to overfitting on the novel classes, resulting in a well-known catastrophic forgetting issue.

Some prior works have linked the problem of catastrophic forgetting with high adaptability (or plasticity) during incremental training (Chi et al., 2022; Zhao et al., 2021). Consequently, these approaches reduce the adaptability by utilizing an incremental-frozen framework, where the encoder is trained only in the base session and remains frozen during the incremental sessions (Zhao et al., 2021; Tao et al., 2020; Chi et al., 2022; Cheraghian et al., 2021a). The resulting methods provide very high stability, but very little adaptability. This phenomenon is often referred to as the stability-adaptability dilemma (Peng et al., 2022b; Zhao et al., 2023) in the FSCIL, where high stability causes reduced adaptability and vice-versa.

In this work, we combine a collection of techniques under a bag of tricks framework with the goal of concurrently enhancing both the stability and adaptability of FSCIL. These tricks have never been explored

---

*This work was partially done when Shuvendu Roy was an Intern at Google Research.
†This work was partially done when Ali Etemad was a Visiting Researcher at Google Research.

together, and their impact on stability, adaptability, and overall performance has not been studied under the same framework. We categorize the techniques into three main groups: (*i*) stability, (*ii*) adaptability, and (*iii*) training. Stability tricks improve the separation among 'base' classes within the learned embedding space, which minimizes interference with previously learned classes when introducing novel classes during the incremental session. This results in greater stability and reduced forgetting of already learned classes. These approaches include incorporating supervised contrastive loss (Khosla et al., 2020), pre-assigning prototypes (Yang et al., 2022), and including pseudo-classes (Lee et al., 2020) during training. On the other hand, adaptability tricks enhance the model's ability to learn novel classes in the incremental session with careful tuning of the encoder through incremental SubNet tuning (Kang et al., 2023). Finally, training tricks include a pre-training step (Chen et al., 2020), and adding a pre-text task (Gidaris et al., 2018) in order to help boost the overall performance without compromising either stability or adaptability. In summary, the key innovation of our framework is combining a set of modules (tricks), which improve stability, adaptability, and overall performance at the same time. While we did not propose any new components, no prior work has explored such modules (tricks) together in the context of FSCIL under a unified framework.

Extensive experiments on three popular benchmark datasets, namely CIFAR100 (Krizhevsky et al., 2009), *mini*ImageNet (Russakovsky et al., 2015) and CUB200 (Wah et al., 2011), demonstrate the effectiveness of our framework. Overall, the proposed framework improves the performance by 3.22%, 1.1%, and 2.0% on CIFAR-100, CUB-200, and miniImageNet, respectively. Incorporating these tricks not only surpasses the overall accuracy of existing FSCIL methods but also demonstrates improved stability (reduced forgetting), higher adaptability (improvements on novel classes), and better separation in the embedding space. Study on the representations learned under stability tricks demonstrates lower intra-class variance and higher inter-class distance between the learned classes, which ensures lower forgetting and thus higher training stability and overall accuracy. Further study on adaptability tricks shows substantial improvement in the performance on new classes while retaining performance for the learned ones. We also present an ablation study to understand the impact of stability, adaptability, and training tricks on overall performance. This study highlights that stability tricks have the most significant impact, followed by adaptability tricks. Following prior works, we report the main results with 5-shot setting experiments on commonly used benchmark datasets. Moreover, to study the behaviour of our method under data-scarce scenarios, we also experiment with 1-shot and 2-shot settings. Additionally, our framework shows similar improvement for larger encoders like ViT, and larger ResNets. Finally, to evaluate scalability with respect to a number of classes, we also report our results on the large-scale ImageNet-1K (Russakovsky et al., 2009) dataset with 1000 classes. Overall, we make the following contributions:

- We present a new framework for FSCIL that combines a bag of tricks to simultaneously improve the stability, adaptability and overall performance of the model.

- We conduct a thorough analysis of the behaviour and effectiveness of all tricks towards stability and adaptability in the context of FSCIL.

- We significantly improve upon the state-of-the-art methods on FSCIL and establish a robust baseline for future research in this area, including new evaluations on low-shots and with a large number of classes.

## 2 Related Works

### 2.1 Class Incremental Learning

Class-incremental learning (CIL) is a continual learning paradigm that involves continuously learning novel classes while preserving the knowledge of previously learned ones (Masana et al., 2022). Existing literature on CIL can be broadly categorized into three main groups. The first group, commonly referred to as the replay-based method, stores past samples in a memory bank for rehearsal during incremental sessions to ensure retention of old knowledge (Rebuffi et al., 2017; Rolnick et al., 2019). The second group is regularization-based, focusing on preventing significant changes in the parameters to prevent forgetting (Li & Hoiem, 2017; Liu et al., 2018). In contrast, the third group dynamically expands the network architecture to accommodate the

learning of novel classes (Zhu et al., 2021a; Shi et al., 2022). In practice, the model is trained over several sessions, with each session introducing novel classes to learn. In traditional CIL, each session includes an adequate number of labelled samples for every novel class, as well as the option to store some of the previous samples for rehearsal in future incremental sessions. However, as per the definition and problem setup of FSCIL, neither of these assumptions holds.

## 2.2 Few-Shot Class-Incremental Learning

In real-world applications, assuming that an incremental session contains a large number of samples for each novel class is impractical (Tao et al., 2020). FSCIL (Tao et al., 2020; Hersche et al., 2022; Cheraghian et al., 2021b; Liu et al., 2022) tackles this challenging scenario where the model needs to incrementally learn novel classes with only a few samples per class. It also assumes that no samples from previous sessions are available, which causes privacy concerns in many domains. Consequently, none of the traditional CIL methods, mentioned in the previous section, perform well in the FSCIL setting (Zhou et al., 2022a; Tao et al., 2020). However, there have been developments in FSCIL literature, which can be discussed in two broad categories. The first category trains the model only on the base session, keeping the model frozen in the incremental session (Zhu et al., 2021b; Shi et al., 2021; Zhang et al., 2021); while the second group tunes the model in the incremental sessions (Tao et al., 2020; Cheraghian et al., 2021a; Dong et al., 2021; Zhao et al., 2021).

The main idea of the first category (frozen encoder-based methods) is to ensure greater separability of base classes in the embedding space so that novel classes in the incremental step can easily fit in this space with minimal interference (Zhu et al., 2021b; Shi et al., 2021; Zhang et al., 2021). This is often referred to as forward compatibility (Song et al., 2023). Prior works in the literature proposed different techniques aiming to enable forward compatibility. For instance, FACT (Zhou et al., 2022a) proposed to use virtual prototypes to force the embedding of different classes to be maximally separated while respecting the relative positioning of classes in the embedding space. SAVC (Song et al., 2023) created virtual classes during the base session, and (Yang et al., 2022) assigned random prototypes that are maximally separated from one another. These methods offer relatively high stability, but little adaptability for incremental training.

The second group of methods tunes the encoder in the incremental session to provide better flexibility for learning new tasks, thus providing better adaptability. For example, MgSvF (Zhao et al., 2021) strategically updated different components at different rates, effectively balancing the adaptation to new knowledge and the preservation of old ones. The exemplar relation distillation framework (Dong et al., 2021) constructed and updated an exemplar relation graph to facilitate the learning of novel classes. SoftNet (Kang et al., 2023) proposed to utilize the concept of the lottery-ticket hypothesis to find a sub-network of important parameters from the previous session, which is left frozen during incremental tuning with the rest of the parameters. A common problem of this second group of methods is that adaptability comes at the cost of stability. That is, the model's performance in the old classes deteriorates as it learns novel classes. Overall, there is a lack of balance between the stability and adaptability in existing FSCIL methods, which we aim to improve with our bag of tricks.

## 3 Method

### 3.1 Overview

In FSCIL, a model is trained across $T$ consecutive sessions, with each session introducing novel classes for the model to learn. Training data for each session $t \in T$ is labelled, $\mathcal{D}_{train}^{t} = \{(x_i,\ y_i)\}_{i=0}^{N_t}$, where $x_i$ and $y_i$ are the $i$-th sample and the corresponding label. In FSCIL, only the base session (first session) contains a sufficient amount of samples for effective training. Subsequent incremental sessions contain only a few samples per class, typically organized in an $N$-way $K$-shot format, containing $K$ training samples for each of the $N$ classes. By definition, each session exclusively contains samples from novel classes, meaning that the label space for each session ($\mathcal{C}^t$) is mutually exclusive with others. The performance of a method is evaluated after each session on the test set $\mathcal{D}_{test}^t$, which contains samples from all classes encountered so far. In our

framework, we include a pre-training stage before the base training. Overall, the framework is divided into three training stages: pre-training, base training, and incremental training.

## 3.2 Baseline

We consider the incremental frozen framework as our baseline due to its proven effectiveness (Zhang et al., 2021; Shi et al., 2021; Song et al., 2023) in addressing the data-scarce incremental learning scenario of FSCIL. In this framework, the model $\phi(x)$ is trained only on the base session ($\mathcal{D}^0_{train}$) using a standard cross-entropy loss, $L_{ce} = (\phi(x),\ y)$. The model $\phi(x)$ consists of an encoder, $f_\theta(x) \in \mathbb{R}^{d \times 1}$ and a classifier head $W \in \mathbb{R}^{d \times |\mathcal{C}^0|}$. The prediction can therefore be expressed as $\phi(x) = W^T f_\theta(x)$. After training on the base session, the encoder $f(x)$ remains frozen during the incremental sessions. To classify novel classes, the classifier $W$ is expanded with the classifier weight for novel classes parameterized by the prototype of each class. A prototype is the average of the embeddings of all samples belonging to that class, $w_c^t = \frac{1}{n_c^t}\sum_{i=1}^{n_c^t} f_\theta(x_{c,i})$.

## 3.3 Stability Tricks

Stability tricks in our framework revolve around the idea that better separation of 'base' classes ensures improved stability in learning new classes in the incremental sessions (Song et al., 2023). As illustrated in Figure 1, better separation of base classes in the embedding space allows the novel classes to be placed in the embedding

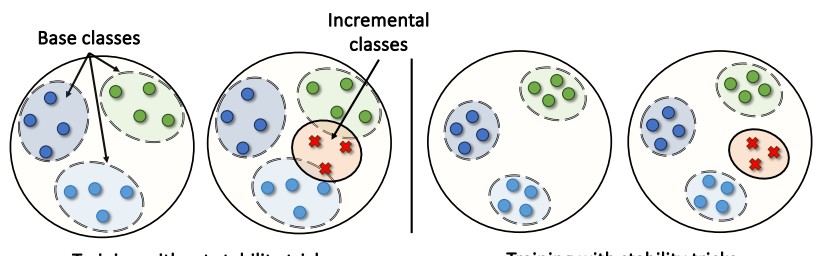

Figure 1: The intuition behind stability tricks. Better separation of base classes ensures stability in incremental learning.

space without interfering with existing classes. This involves increasing the distance between classes (inter-class distance) while reducing the distance between samples within the same class (intra-class distance). This approach is also known as forward compatibility in FSCIL literature. Accordingly, we incorporate three techniques that can effectively promote better stability: training with a supervised contrastive loss, pre-assigning prototypes, and including pseudo-classes.

**Supervised Contrastive Loss.** While most of the existing literature on FSCIL use the standard cross-entropy loss for learning during the base session, some prior works (Song et al., 2023) have demonstrated that cross-entropy does not effectively separate classes in the embedding space. Some studies have indicated that the supervised contrastive loss (SupCon) (Khosla et al., 2020) exhibits better separability in the embedding space. SupCon is a variation of the popular contrastive loss (Chen et al., 2020), which additionally includes class labels to guide representation learning in a supervised manner. Specifically, SupCon learns representations by pulling the samples (and their augmentations) of the same class closer in the embedding space while pushing the samples of different classes apart, resulting in a more separable embedding space than cross-entropy. In other words, SupCon forces the representation of each sample to be close to its corresponding class prototype (center of embedding), while pushing the prototypes away from one another. For a batch of labelled samples $\{(x_i,\ y_i)\}_{i=0}^N$, the SupCon loss can be represented as:

$$\mathcal{L}_{sup} = \sum_{i=1}^{N}\left(\frac{-1}{N_{y_i}-1}\sum_{j=1}^{N}\mathbf{1}_{[i \neq j]}\cdot\mathbf{1}_{[y_i = y_j]}\cdot log\frac{exp(z_i \cdot z_j/\tau)}{\sum_{k=1}^{N}\mathbf{1}_{[k \neq i]}exp(z_i \cdot z_k/\tau)}\right)\ , \tag{1}$$

where $z = f_\theta(x)$, $N$ is the batch-size, and $Ny_i$ is the number of positive samples from class $y_i$. The indicator function denoted by $\mathbf{1}_{[y_i = y_j]}$ yields a value of 1 when indices $i$ and $j$ correspond to instances of the same class, $\tau$ is a temperature parameter.

**Pre-assigning Prototypes.** In SupCon, prototypes are learned along with the optimization of the model. However, it was shown in Yang et al. (2022) that 'pre-assigning' prototypes that are maximally separated

from one another ensures improved separation in the embedding space. In Yang et al. (2022), the maximally separated prototypes were defined as simplex Equiangular Tight Frame (ETF) (Papyan et al., 2020) $P$, a geometric arrangement of $K$ vectors in $d$-dimensional Euclidean space (Yang et al., 2022). Each vector in this space has an Euclidean norm of 1, with any pair of distinct vectors yielding an inner product of $-1/(K-1)$. This specific inner product value corresponds to the largest possible angle between any two vectors in this space. Our implementation of the prototype pre-assignment slightly differs from (Yang et al., 2022) in two ways. First, we do not rely solely on the cross-entropy loss but instead incorporate the SupCon loss introduced earlier. Second, we do not assign the ETF prototypes randomly at the very beginning of training. Instead, we train the encoder with the SupCon for a few epochs and take advantage of the prototypes naturally formed by SupCon. Then, for each class, we assign the ETF prototype that is most closely aligned with its naturally formed prototype, and train the encoder to minimize the distance between the learned and pre-assigned prototype as:

$$\mathcal{L}_{ETF} = \frac{1}{C^0} \sum_{i=1}^{C^0} (P_c - w_c)^2, \tag{2}$$

where, $P_c$ and $w_c$ are the assigned and learned prototypes of class $c$, and $C^0$ is the total number of classes in the base session.

**Including Pseudo-classes.** Previous studies such as FACT (Zhou et al., 2022a) and SAVC (Song et al., 2023) introduced the concept of integrating pseudo-classes during the base session to serve as placeholders in the embedding space for novel classes. In SAVC, the pseudo-classes were generated by a pre-defined transformation, which was considered a more fine-grained variant of the original classes. In our work, we take a similar approach of including pseudo-classes from (Lee et al., 2020), which employs hard augmentations to transform the semantics of the sample and consider it as a pseudo-class. Let, $\mathcal{F}$ be a set of pre-defined (hard) augmentations for pseudo-class formation, and $x_{c,i}$ be a sample of class $c$. With the pseudo-class trick, we consider the transformation of the image $\mathcal{F}(x_{c,i})$ as an instance of a pseudo-class $(C^0 * M + c)$, where $C^0$ is the total number of classes in base session, and $M$ is the pseudo-class multiplication factor. Such pseudo-classes can be seen as a fine-grained class derived from the original class. In our work, we use $M = 2$, which doubles the total number of classes (including pseudo-classes) during base training. This trick does not include any new loss function.

### 3.4 Adaptability Tricks

While stability tricks help the model retain the knowledge of base classes, they offer limited adaptability for the model to learn novel classes effectively. Independent performance evaluation of base and novel classes at the end of training demonstrates that the performance of novel classes is substantially inferior to that of base classes (Song et al., 2023). Consequently, the overall performance at the end of training is predominantly influenced by the performance of the base classes. This underscores the need for FSCIL methods to enhance the model's adaptability on incremental sessions to improve the model's performance on novel classes. In this section, we discuss two tricks that provide more adaptability for the model: incremental fine-tuning and SubNet tuning, which we combinedly denote as Incremental SubNet Tuning.

**Incremental SubNet Tuning.** Fine-tuning is a common practice in machine learning literature to tune a pre-trained model for a new task or setting. It is also a widely used technique across conventional continual learning literature, predominantly with rehearsal-based techniques. However, fine-tuning in the context of FSCIL requires careful consideration of the training setup, since no data from the previous session is available for rehearsal during the current session. Consequently, tuning can cause catastrophic forgetting of already learned knowledge. In our framework, we adopt the fine-tuning concept of Song et al. (2023), which utilizes a small learning rate to tune certain portions of the pre-trained encoder, while keeping the rest of the encoder frozen. Specifically, we freeze the shallow layers of the network since shallower layers are known to capture domain-invariant features, whereas the deeper layers learn more fine-grained features.

While such an incremental fine-tuning approach provides more adaptability for learning novel classes, it may result in decreased stability. To deal with this issue, we combine this adaptability trick with the concept of

SubNet tuning, which maintains the adaptability of incremental fine-tuning without a substantial drop in stability. The concept is inspired by the Lottery-ticket hypothesis (Frankle & Carbin, 2018), which states that there exists a subnetwork of a dense network that performs as well as the whole network. In Kang et al. (2023), this concept is utilized in the context of FSCIL to find a sub-network of the trained encoder that performs on par with the whole network on the base classes. Adopting this into our framework, after the training of the encoder ($f_\theta$) on the base session, we find a subset of network parameter that performs as well as the whole network on the base session.Specifically, we follow Kang et al. (2023) to extract a SubNet (subnetwork) mask $m^*$ as:

$$m^* = \min_{m \in [0,1]^{|\theta|}} \frac{1}{n} \sum_{i=1}^{n} \mathcal{L}\big(f_{(\theta \odot m)}(x_i), y_i\big) - \mathcal{L}\big(f_\theta(x_i), y_i\big), \tag{3}$$

where $\odot$ is an element-size dot-product operation. Here, $\mathcal{L}$ is the loss function for training the model on the base session, and $m^*$ is the optimal binary mask with the same size as the network, for which the performance of the masked SubNet is comparable to that of the original network. During the incremental session, we freeze the sub-network to ensure the performance of the base session and fine-tune the remaining parameters for learning the novel classes.

### 3.5 Training Tricks

Building upon the principles of stability and adaptability tricks discussed earlier, we introduce a set of training techniques that can further enhance overall performance without compromising either stability or adaptability. These tricks include adding a pre-training step and including an additional learning signal.

**Additional Pre-training Step.** Existing research in this field indicates that self-supervised pre-training, followed by supervised fine-tuning, consistently outperforms fully supervised training, particularly in scenarios with limited training data. In the context of FSCIL where data scarcity is a significant challenge, leveraging a self-supervised pre-training step has the potential to provide substantial benefits. As a result, we introduce a contrastive self-supervised pre-training step before the training of the base session. The contrastive self-supervised loss is similar to the SupCon loss introduced earlier, except no label information is utilized. Accordingly, the contrastive pre-training loss can be represented as:

$$\mathcal{L}_{con} = -\frac{1}{2b} \sum_{i=1}^{2b} log \frac{exp(z_i, z_{\kappa(i)}/\tau)}{\sum_{k=1}^{2b} \mathbf{1}_{[k \neq i]} exp(z_i, z_k/\tau)}, \tag{4}$$

where, $\kappa(i)$ is the index of the second augmented sample, and $\mathbf{1}_{[k \neq i]}$ is an indicator function that returns 1 when $k$ is not equal to $i$, and 0 otherwise. $\tau$ is a temperature parameter, and $b$ is the batch size.

**Including Additional Learning Signal.** Following the intuition from the previous trick, we include another self-supervised learning signal, but this time while training on the base session rather than as a separate step. Existing literature on other data-scarce scenarios, such as semi-supervised learning, has shown that adding a pre-text task along with the supervised learning helps the model learn better representation without overfitting to the small labelled set (Berthelot et al., 2019; Roy & Etemad, 2023). To this end, we include a rotation prediction task (Gidaris et al., 2018) that is shown to perform well with supervised learning (Berthelot et al., 2019). Here, the basic idea is to apply a rotation operation on the input image, and the task is to predict the amount of rotation applied to the image. In practice, a rotation module randomly samples one of the following rotations and applies it to the image: $0°, 90°, 180°, 270°$. As a result, the rotation prediction task can be viewed as a four-way classification task, represented as:

$$\mathcal{L}_{rot} = \mathcal{H}(r, P_{\theta_r}(r|Rotate(x)). \tag{5}$$

Here, $x$ is the input image, $P_{\theta_r}$ is the encoder that predicts the rotation, and $\mathcal{H}$ is the cross-entropy loss.

## 4 Experiments and Results

### 4.1 Datasets and Implementation Details

Following the established protocol in the FSCIL literature, we conduct our experiments on three popular datasets: CIFAR100 (Krizhevsky et al., 2009), *mini*ImageNet (Russakovsky et al., 2015) and CUB200 (Wah et al., 2011). To ensure a fair comparison with prior works (Tao et al., 2020; Chi et al., 2022) on FSCIL, we use the same encoder (ResNet-18), and data split across the training sessions. Specifically, for CIFAR-100 and miniImageNet, we use 60 classes for the base session and 40 classes for the incremental sessions. The incremental learning experiments are conducted on a 5-way, 5-shot setting. In the case of CUB-200, we allocate 100 classes for the base session and another 100 classes for the incremental sessions, each containing ten classes (10-way, 5-shot). Further details on implementation and hyper-parameters are presented in Appendix S1.

### 4.2 Evaluation Protocols

Following the standard evaluation protocol in FSCIL literature (Tao et al., 2020; Song et al., 2023; Kang et al., 2023), we report the model's accuracy after each incremental session. To further understand the properties of the learned representations, we investigate inter-class distance, intra-class distance, and class separation. Following, we define these properties.

**Inter-class distance** is the distance between the prototypes of any two classes in the embedding space. Given, two class prototypes $w_i$ and $w_j$ for classes $i$ and $j$, we compute their inter-class distance as:

$$d_{inter}^{i,j} = 1 - cosine(w_i, w_j), \tag{6}$$

where $cosine()$ is the cosine similarity between two vectors.

**Intra-class distance** indicates the average distance from a prototype to the samples belonging to that class. For class $k$, we compute the intra-class distance as:

$$d_{intra}^k = 1 - \frac{1}{n_k} \sum_{i=1}^{n_k} cosine(z_i, w_k), \tag{7}$$

where $n_k$ is the number of samples belonging to class $k$, and $z_i$ is the embedding of the $i^{\text{th}}$ sample.

**Class-separation** determines how well the samples from one class are separated from other classes in the embedding space. For a dataset with $C$ classes, class separation can be represented as: $1 - d_{within}/d_{total}$. Here, $d_{within}$ is the average distance between samples of the same classes, while $d_{total}$ is the average distance between samples in the embedding space. Accordingly, they are formulated as:

$$d_{within} = \sum_{c=1}^{C} \sum_{i=1}^{n_c} \sum_{j=1}^{n_c} \frac{1 - cosine(z_{c,i}, z_{c,j})}{C \cdot n_c^2}, \tag{8}$$

and

$$d_{total} = \sum_{c=1}^{C} \sum_{d=1}^{C} \sum_{i=1}^{n_c} \sum_{j=1}^{n_d} \frac{1 - cosine(z_{c,i}, z_{d,j})}{C^2 \cdot n_c \cdot n_d}. \tag{9}$$

### 4.3 Main Results

In this section, we present the main results of our tricks by cumulatively adding them to the baseline. The results are presented in Table 1, which include the category of the tricks, along with the stages at which the tricks are applied and the performance on CIFAR-100, CUB-200 and miniIN. As outlined in Section 3.2, we adopt the incremental-frozen framework as the baseline for our study. As shown in Table 1, the accuracy of this baseline for CIFAR-100, CUB-200, and miniImageNet datasets are 43.77%, 59.88%, and 45.08%, respectively. To ensure the best performance for the baseline, we perform an extensive hyper-parameter study

Table 1: Impact of different tricks on the baseline for all datasets. Here, the tricks are added cumulatively to the baseline, and the 'Stage' indicates which stage of training the tricks are applied.

| Category | Trick | Stage | CIFAR-100 | CUB-200 | miniIN |
|---|---|---|---|---|---|
| – | Baseline | – | 43.77 | 59.88 | 45.08 |
| Stability | + *SupCon* | Base+Incremental | 50.16 | 60.38 | 48.90 |
| | + *ETF vector* | Base | 51.10 | 60.74 | 49.73 |
| | + *Pseudo-classes* | Base | 51.65 | 62.18 | 54.79 |
| Adaptability | + *Inc. SubNet tuning* | Incremental | 58.12 | 63.10 | 57.85 |
| Training | + *Pre-training* | Pre-training | 58.34 | 63.30 | 58.01 |
| | + *Additional signal* | Base | 58.55 | 63.60 | 59.11 |

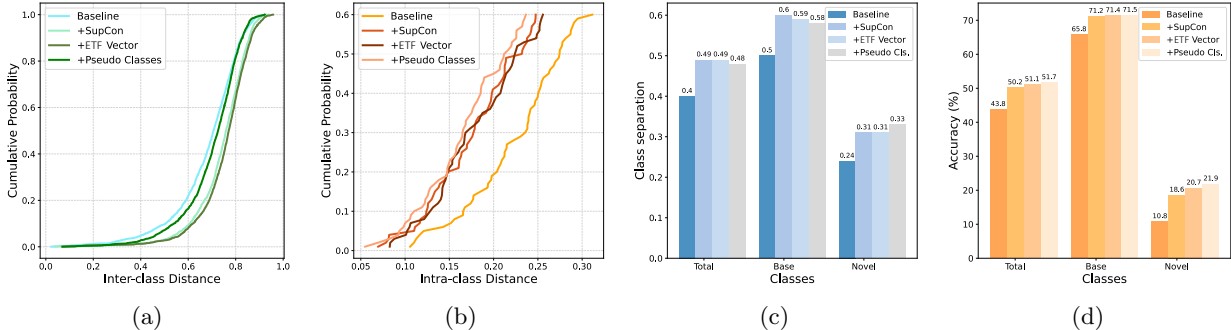

(a)       (b)       (c)       (d)

Figure 2: Properties of stability tricks on CIFAR-100. (a) Presents inter-class distance (the distance between class prototypes), which we aim to maximize for better stability during incremental training; (b) depicts the intra-class distance (the average distance of samples from the corresponding prototypes), which we aim to minimize for better stability; (c) presents the class separation degree (the overall separability of classes ranging between 0 and 1), which we aim to maximize; (d) presents the accuracy of Base, Novel, and Total classes.

presented in Appendix S3.1. As we later compare our results with prior works (Section 4.5), we observe that this baseline outperforms many of the prior studies, showing the robustness of our baseline.

**Stability Tricks.** Next, we discuss the results of including stability tricks into the baseline, beginning with the addition of SupCon loss. The results from this study show substantial improvement for all datasets (Table 1), obtaining accuracies of 50.16%, 60.38%, and 48.90%, for CIFAR-100, CUB-200, and miniImageNet respectively. As discussed earlier, a key element for ensuring high stability in FSCIL is to ensure increased separability in the embedding space (increased inter-class distance and reduced intra-class dispersion). In Figures 2a and 2b, we plot the cumulative probability of inter-class and intra-class distances (defined in Section 4.2) for different tricks. We plot the cumulative probability of inter-class and intra-class distance (Song et al., 2023) instead of the average since the average distance can be misleading due to the presence of outliers, whereas the cumulative probability provides a more robust and nuanced representation of the distribution of distances. As we observe from Figure 2a, adding SupCon provides a large increase in the inter-class distance compared to training with cross-entropy loss in the baseline. At the same time, SupCon greatly reduces the intra-class distance compared to the cross-entropy-based baseline (illustrated in Figure 2b). However, our findings diverge from those of Song et al. (2023), which indicated that while supervised contrastive loss effectively reduces intra-class distances, it unexpectedly leads to a reduction in inter-class distances as well. The divergence in our findings may be attributed to implementation specifics. For example, SAVC utilized multi-crop augmentation, which we do not use. Also, we adopted a different set of hyper-parameters determined through our study (Appendix S3.2).

We also show the overall class separation degree for different tricks in Figure 2c. Class separation degree refers to the degree of distinctiveness or separability between different categories within the embedding space, which can be measured as defined in Section 4.2. It typically ranges between 0 and 1, with higher values indicating clearer boundaries and better separability between classes. As we observe, adding SupCon provides better class separation for both base and novel classes. Overall, the higher separation in the embedding space

contributes to the large improvement in the overall accuracy. Analyzing the performance of both base and novel classes in Figure 2d, we find improvements for both, with a 5.4% boost in base class accuracy and a 7.8% increase in novel class accuracy.

Next, we include our second stability trick of pre-assigning ETF vectors as prototypes. As we see in Figure 2a, this trick further increases inter-class distance (Figure 2a) over the previous trick. Although this trick does not further reduce the intra-class distance (Figure 2b), it increases the class separation of novel classes (Figure 2c), resulting in a 2.1% improvement in novel classes. Overall, including this trick increases the final performance to 51.10%, 60.74%, and 49.73%, respectively.

Finally, adding pseudo-classes causes a further reduction in the intra-class distance since it requires fitting twice the number of classes in the same amount of space. Although this trick reduces the inter-class distance, it provides an increase in the novel class separation. This results in a 1.20% improvement in novel classes and 0.6% in the final performance. Overall, we find an improved accuracy of 51.21%, 62.27%, and 55.82% for CIFAR-100, CUB-200, and miniImageNet, respectively. Here, we find the largest improvement of 5.06% on the miniImageNet dataset.

**Adaptability Tricks** As discussed earlier, stability tricks do not provide sufficient adaptability for the model to perform well on novel classes. This is evident from the accuracy on base and novel classes in Figure 2d, where we see a 49.6% difference in the performance on base and novel classes. Specifically, the performance on base and novel classes are 71.5% and 21.9%, respectively. This indicates that the model struggles to effectively learn the novel classes, demonstrating the need for adaptability tricks.

Our adaptability trick with Incremental SubNet Tuning provides adaptability by means of fine-tuning a sub-network on the encoder in incremental sessions. As shown in Figure 3a, this trick improves accuracy on novel classes by 11.1%. We also show the accuracy after each session in Figure 3b. From this figure, we observe comparable accuracies to that of stability tricks in the earlier sessions. However, Incremental SubNet Tuning shows better performance in later sessions when the number of novel classes increases. This again shows the importance of the adaptability trick for learning novel classes. Overall, this trick provides an improved accuracy of 58.12%, 63.1%, and 57.85% on the CIFAR-100, CUB-200, and miniImageNet.

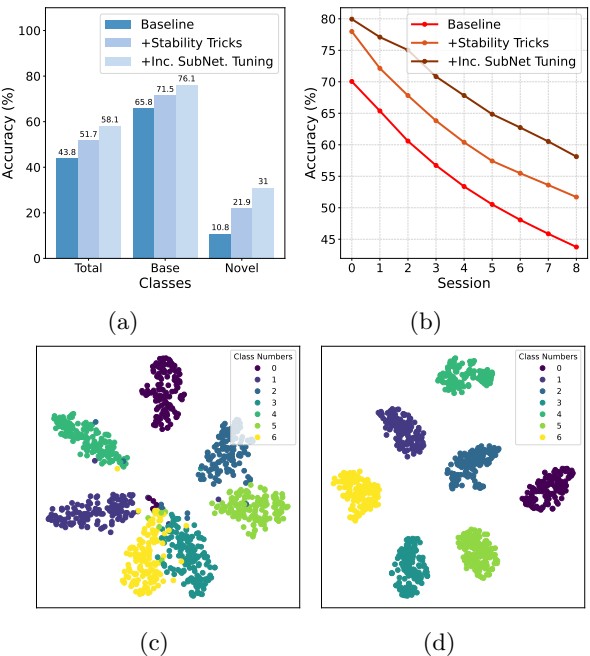

(a)  (b)

(c)  (d)

Figure 3: Properties of adaptability tricks on CIFAR-100. (a) Presents accuracy of Base, Novel, and Total classes; (b) presents the total accuracies after each session, which we aim to maximize; (c) and (d) depict t-SNE visualizations for stability and adaptability tricks, where incorporating adaptability tricks shows more separation. Here, 0-4 are base classes, and 5-6 are novel classes.

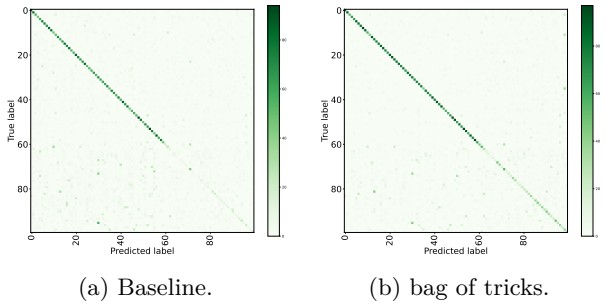

(a) Baseline.  (b) bag of tricks.

Figure 4: Confusion matrices for the baseline and our bag of tricks. The baseline performs well on the base session, but performance drops for novel classes. Our framework shows improved performance for both base and novel classes.

In Figure 3c, we visualize a t-SNE representation of the learned embeddings (for a randomly selected subset of classes) for the stability tricks and the adaptability tricks. As observed in these figures, stability tricks alone provide sufficient space for the integration of novel classes (5 and 6), although it results in a slight overlap with base classes. In contrast, the incorporation of adaptability tricks creates more separable embeddings.

**Training Tricks.** Finally, we discuss the results of adding the training tricks. First, we investigate the impact of including the pre-training trick. With the pre-training trick with self-supervised learning, the performance on the three datasets is increased to 58.34%, 63.30%, and 58.01%, respectively. Finally, we observe additional improvements with our last training trick of incorporating the additional learning signal, resulting in accuracies of 58.55%, 63.60%, and 59.11%, on CIFAR-100, CUB-200, and miniImageNet, respectively. To provide an overall evaluation, we present the confusion matrices for both the baseline and the complete bag of tricks in Figure 4. For the baseline, noticeable dark points (correct predictions) are observed along the diagonal for the base classes (first 60 classes), but these become less prominent for the incremental classes. With the bag of tricks, we observe significant improvements for all classes, including novel ones.

### 4.4 Ablation Study

In this section, we present an ablation study on the stability, adaptability and training tricks on the CIFAR-100 dataset. The results of this study are presented in Table 2. It is evident from the table that stability tricks have the highest individual impact on the performance of the model, removing which results in an 11.2% drop in performance. Furthermore, removing stability tricks with any of the other tricks drastically drops the performance, resulting in a final accuracy worse than the baseline. This is caused by the fact that without stability tricks, adaptability tricks alone overfit the new classes, and training tricks alone do not learn generalized features. We find the second important component to be the adaptability tricks, which result in a 2.7% drop in performance when removed.

Table 2: Ablation study.

| Stab. | Adap. | Train. | Acc |
|:-----:|:-----:|:------:|:-----:|
| ✓ | ✓ | ✓ | 58.55 |
| ✓ | ✓ | ✗ | 58.12 |
| ✓ | ✗ | ✓ | 55.85 |
| ✗ | ✓ | ✓ | 47.35 |
| ✓ | ✗ | ✗ | 51.65 |
| ✗ | ✓ | ✗ | 37.34 |
| ✗ | ✗ | ✓ | 40.19 |
| ✗ | ✗ | ✗ | 43.77 |

**Performance with Larger Encoders.** For all the experiments in this work, we followed the same problem setup and encoders used by all the prior works on FSCIL (Song et al., 2023; Kang et al., 2023). Our focus in this paper is to improve the stability, adaptability, and overall performance of FSCIL while following the same benchmarks and problem setups, as well as encoder backbones, to ensure a fair comparison. However, the tricks in our framework are independent of the encoder's choice and can be easily adapted to any encoder, including ViT. To investigate the performance of our framework on larger encoders, in Table 3, we show the results on ResNet-50, ViT-B/16, and pre-trained (on ImageNet) ViT-B/16, ViT-B/32, and ViT-L/14. The results from this study show that the proposed tricks not only transfer to larger encoders and ViTs, but the performance gains are even larger than those of smaller networks like ResNet-20. For

Table 3: Performance of prior works and our framework for different encoder sizes on CIFAR-100 dataset.

| Encoder | Accuracy | |
|---------|:--------:|:----:|
|  | Baseline | Ours |
| ResNet-20 | 43.77 | 58.55 |
| ResNet-50 | 38.01 | 58.75 |
| ViT-B/16 | 32.14 | 47.89 |
| ViT-B/16 (pre-trained) | 50.14 | 69.72 |
| ViT-B/32 (pre-trained) | 50.01 | 69.58 |
| ViT-L/14 (pre-trained) | 51.55 | 71.09 |

instance, ResNet-50 shows a 20.74% improvement compared to 14.78% in ResNet-20. Similarly, ViT-B/16 with pre-trained and randomly initialized encoders shows 19.58% and 15.75% improvements over the baseline. Additionally, pre-trained ViT-B/32 and ViT-L/14 show 19.57% and 19.54% improvements over the baseline. This is due to the fact that without the stability and adaptability tricks from our framework, a larger encoder with more parameters is more prone to overfitting, making it difficult to learn incremental classes in a few-shot learning setting. With our proposed framework, the larger encoders show a large improvement in performance.

### 4.5 Comparison to Prior Works

Finally, we compare the results of our bag of tricks with the prior works on CIFAR-100 in Table 4, and present the results for CUB-200 and miniImageNet in Appendix Tables S1 and S2. The tables present the

Table 4: Comparison to prior works across the base and incremental sessions on CIFAR-100 for 5-way, 5-shot setting. **The results for CUB-200 and miniImageNet are presented in the Appendix.**

| Method | Acc. in each session (%) ↑ | | | | | | | | |
|---|---|---|---|---|---|---|---|---|---|
| | 0 | 1 | 2 | 3 | 4 | 5 | 6 | 7 | 8 |
| Baseline | 70.05 | 65.37 | 60.59 | 56.72 | 53.38 | 50.53 | 48.07 | 45.87 | 43.77 |
| Rebalancing (Hou et al., 2019) | 61.31 | 47.80 | 39.31 | 31.91 | 25.68 | 21.35 | 18.67 | 17.24 | 14.17 |
| iCaRL (Rebuffi et al., 2017) | 61.31 | 46.32 | 42.94 | 37.63 | 30.49 | 24.00 | 20.89 | 18.80 | 17.21 |
| TOPIC (Tao et al., 2020) | 61.31 | 50.09 | 45.17 | 41.16 | 37.48 | 35.52 | 32.19 | 29.46 | 24.42 |
| IDLVQ-C (Chen & Lee, 2020) | 64.77 | 59.87 | 55.93 | 52.62 | 49.88 | 47.55 | 44.83 | 43.14 | 41.84 |
| FSLL (Mazumder et al., 2021) | 66.48 | 61.75 | 58.16 | 54.16 | 51.10 | 48.53 | 46.54 | 44.20 | 42.28 |
| FSLL+SS (Mazumder et al., 2021) | 68.85 | 63.14 | 59.24 | 55.23 | 52.24 | 49.65 | 47.74 | 45.23 | 43.92 |
| F2M (Shi et al., 2021) | 67.28 | 63.80 | 60.38 | 57.06 | 54.08 | 51.39 | 48.82 | 46.58 | 44.65 |
| CEC (Zhang et al., 2021) | 73.07 | 68.88 | 65.26 | 61.19 | 58.09 | 55.57 | 53.22 | 51.34 | 49.14 |
| MetaFSCIL (Chi et al., 2022) | 74.50 | 70.10 | 66.84 | 62.77 | 59.48 | 56.52 | 54.36 | 52.56 | 49.97 |
| CLOM (Zou et al., 2022) | 74.20 | 69.83 | 66.17 | 62.39 | 59.26 | 56.48 | 54.36 | 52.16 | 50.25 |
| C-FSCIL (Hersche et al., 2022) | 77.47 | 72.40 | 67.47 | 63.25 | 59.84 | 56.95 | 54.42 | 52.47 | 50.47 |
| LIMIT (Zhou et al., 2022b) | 73.81 | 72.09 | 67.87 | 63.89 | 60.70 | 57.77 | 55.67 | 53.52 | 51.23 |
| FACT (Zhou et al., 2022a) | 74.60 | 72.09 | 67.56 | 63.52 | 61.38 | 58.36 | 56.28 | 54.24 | 52.10 |
| SAVC (Song et al., 2023) | 79.85 | 73.70 | 69.37 | 65.28 | 61.91 | 59.27 | 57.24 | 54.97 | 53.12 |
| ALICE (Peng et al., 2022a) | 79.00 | 70.50 | 67.10 | 63.40 | 61.20 | 59.20 | 58.10 | 56.30 | 54.10 |
| SoftNet (Kang et al., 2023) | 79.88 | 75.54 | 71.64 | 67.47 | 64.45 | 61.09 | 59.07 | 57.29 | 55.33 |
| **Ours** | **80.25** | **77.20** | **75.09** | **70.82** | **67.83** | **64.86** | **62.73** | **60.52** | **58.55** |

*We reproduced the results for SAVC as the original paper does not provide the exact values for each session.

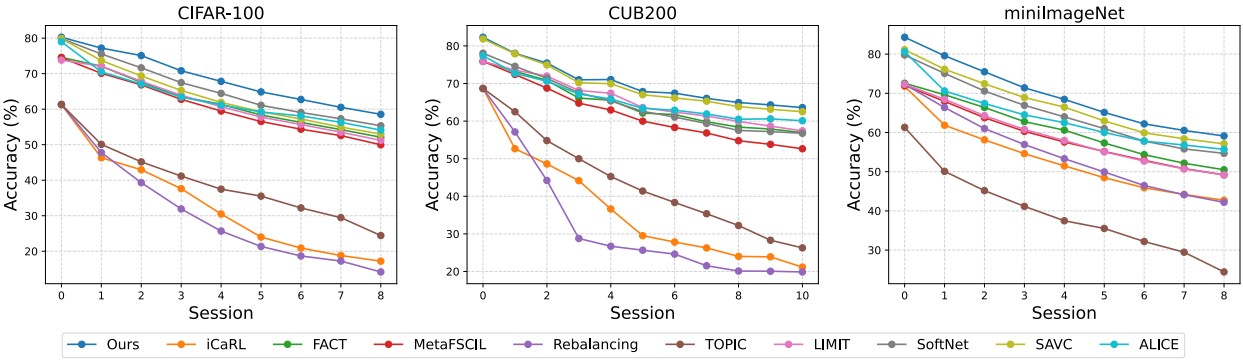

Figure 5: Comparison to prior works across CIFAR-100, CUB-200, and miniImageNet datasets, demonstrating that our solution outperforms prior works.

results for the base session and the overall performance across incremental sessions. The first row in this table presents the results for our baseline. Among prior works, SoftNet held the previous state-of-the-art for CIFAR-100 with a final accuracy of 55.33%. Our framework achieves a performance of 58.55%, marking a 3.22% improvement over the current state-of-the-art. In Figure 5, we plot the accuracy over the incremental sessions for CIFAR-100, CUB-200, and miniImageNet. For CUB-200, the previous state-of-the-art was held by SAVC with an accuracy of 62.50%, which our framework outperforms with an accuracy of 63.60%. Finally, the state-of-the-art on miniImageNet was also held by SAVC, with an accuracy of 57.11%, which our method outperforms by 2.0%, achieving an accuracy of 59.11%.

**Performance Across Different Shots.** Following prior works in FSCIL (Zhou et al., 2022a; Tao et al., 2020), we report the main results in a 5-shot setting. Nonetheless, We also investigate the performance on 1-shot and 2-shot settings to evaluate how the model performs in data-scarce settings and compare the performance with two of the previous state-of-the-art methods, SAVC and SoftNet. We also investigate the 10-shot setting to determine how an increase in data impacts performance. As seen from Figure 6, in 1-shot learning, our proposed framework outperforms prior works by a larger margin than the 5-shot setting. This

Table 5: Performance on the ImageNet-1K dataset. We use 500 classes as base sessions and perform 50-way, 10-shot incremental learning.

| Method | Acc. in each session (%) ↑ | | | | | | | | | | |
|---|---|---|---|---|---|---|---|---|---|---|---|
| | 0 | 1 | 2 | 3 | 4 | 5 | 6 | 7 | 8 | 9 | 10 |
| SAVC (Song et al., 2023) | 37.74 | 35.94 | 34.45 | 33.16 | 31.88 | 30.93 | 29.80 | 28.83 | 28.1 | 27.37 | 26.12 |
| SoftNet (Kang et al., 2023) | 36.21 | 34.23 | 32.35 | 30.53 | 28.71 | 27.15 | 25.68 | 24.31 | 23.16 | 22.19 | 21.21 |
| **Ours** | 39.34 | 37.86 | 36.49 | 35.23 | 34.30 | 33.83 | 32.29 | 31.21 | 30.77 | 29.34 | 28.25 |

further shows the effectiveness of the bag of tricks in data-scarce settings. More experiments on 1-, 2-, 5- and 10-shot settings are discussed in Appendix S3.5.

**Performance on ImageNet-1K.** So far we have report the main results of our approach on, CIFAR-100, CUB-200, and miniImageNet, which are commonly used by prior works in this area. However, these datasets are relatively small in terms of the number of classes. To better understand the scalability of FSCIL in terms of the number of classes, we conduct an experiment on the ImageNet-1K (Russakovsky et al., 2009) dataset, which contains 1000 classes. For this study, we consider 500 randomly selected classes as the base classes and report the results for a 50-way, 10-shot incremental learning setting. The results of this study with our method and its comparison to prior methods are presented in Table 5. As seen from the results, our proposed framework outperforms prior works on the ImageNet-1K dataset by a considerable margin. More specifically, our framework achieves 7.04% and 2.13% improvement over SoftNet and SAVC, respectively.

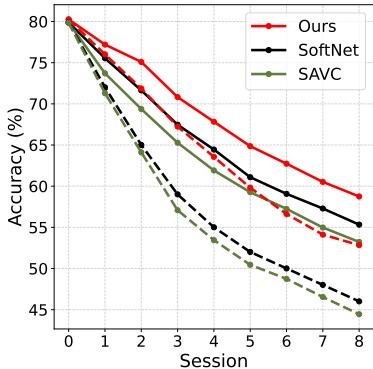

Figure 6: Performance for different shots. Solid and dotted lines represent 5- and 1-shot performances.

### 4.6 Time Complexity

In Table 6, we discuss the time complexity of our framework using a single Nvidia RTX 2080 GPU in comparison to SAVC (Song et al., 2023). To this end, we report the throughput in frames per second (FPS) and the training time. During the training phase on the base session, the throughputs for SAVC and our framework are 630 FPS and 450 FPS, respectively. Nonetheless, during incremental training and inference, our framework with ResNet-18 is as fast as SAVC. Con-

Table 6: Comparison of computational complexity with a batch-size of 64.

| Method | Acc. | Train | | Infer. | Runtime. |
|---|---|---|---|---|---|
| | | base | inc. | | |
| SAVC | 53.12 | 810 | 950 | 2150 | 6.1h |
| Ours (ResNet-18) | 58.55 | 630 | 935 | 2150 | 6.8h |
| Ours (ResNet-50) | 58.75 | 450 | 710 | 1800 | 8.1h |

sequently, once trained, our framework can be reliably deployed in real-world applications with the same inference time as previous state-of-the-art while achieving enhanced performance.

### 4.7 Discussion

According to the setup adopted in the literature for FSCIL Song et al. (2023); Kang et al. (2023), each of the new classes in the incremental session is different from the classes seen so far. Therefore, incremental classes are generally not exactly identical to the base classes. However, some of the incremental classes can be *similar* to the base classes. For instance, in CIFAR-100, 'Orchid' is a class of flower that is part of the base class, while 'Sun-flower' is another flower that is part of the incremental class. A few similar pairs of examples of base classes with similarities to incremental classes

Table 7: Performance comparison for base and incremental classes with high semantic similarity.

| Method | Base | Incremental |
|---|---|---|
| Baseline | 65.3% | 19.7% |
| Ours | 74.5% | 30.4% |

are 'Lion'-'Tigar', 'Bus'-'Streetcar', and 'Shark'-'Dolphin', respectively. To understand the effectiveness of the stability on *closely related classes*, we report the average accuracy on the above-mentioned base and incremental classes with and without our tricks below in Table 7. As we find from this experiment, even for semantically similar incremental classes, our proposed framework is capable of improving the performance of both the base classes (improved stability) and the incremental ones (improved adaptability).

## 5  Conclusion and Future Work

We present a bag of tricks framework that combines six effective tricks in three distinct categories to improve the stability, adaptability, and overall performance of FSCIL. Stability tricks improve the separation among learned classes to facilitate the learning of new ones, resulting in large improvements on both the base and total classes. Adaptability tricks improve the performance on novel classes by providing more learning capability during incremental sessions. Finally, training tricks provide an additional boost to the final performance. While we did not introduce any new tricks, our contribution in this work lies in developing a framework that combines a set of tricks that jointly improve both stability and adaptability, which, as discussed in the literature review, is a difficult task since improved stability or adaptability often hampers the other aspect. Additionally, We provide extensive analysis of these tricks for a better understanding of their impact beyond the final accuracy, including the impact on stability and adaptability, class separation in the learned embedding space, and performance improvement on base and novel classes. Furthermore, we present a detailed study and comparison to prior works on different aspects that are not explored in the existing literature, including low-shot (1-, 2-shot) performance, results on a dataset with a large number of classes (InageNet-1k), performance on larger encoders (e.g. ResNet-50, ResNet-101), performance on fine-tuning pre-trained foundation models (ViT-B/16, ViT-B/32, and ViT-H/14), performance when new-class has similarity to a base classes (included in revised manuscript). We believe the unified framework and the extensive set of experiments will add value to the further development of the challenging setting of FSCIL.

Though our framework provides adequate adaptability, the performance of the novel classes is still relatively lower compared to the base classes. While this phenomenon has also been widely reported in prior works, our framework was successfully able to reduce the gap between the performance of the base and incremental class from 55% in the literature to 45.2%. However, the performance of the incremental class is still considerably lower than the base class, which can be a potentially interesting direction of inquiry for future work.

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

## Supplementary Material

In this section, we present more details on the implementation and discuss additional experimental results. Following is an overview of the organization of the supplementary material.

## S1  Datasets and Implementation Details

Following previous works (Tao et al., 2020; Song et al., 2023), we evaluate our framework on three popular datasets: CIFAR100 (Krizhevsky et al., 2009), *mini*ImageNet (Russakovsky et al., 2015), and CUB200 (Wah et al., 2011). CIFAR-100 is a dataset of 100 classes, where we use 60 classes (following (Tao et al., 2020)) in the base session and the remaining 40 classes in the incremental sessions. Each incremental session is formulated as a 5-way, 5-shot problem. CUB-200 is a dataset with 200 fine-grained categories. For this dataset, we use 100 classes (following (Tao et al., 2020)) in the base session and the remaining 100 classes in the incremental sessions, with 10 classes in each session. Finally, miniImageNet is a subset of the popular ImageNet dataset that contains 100 classes. For this dataset, we use 60 classes (following (Tao et al., 2020)) for the base session and 5 classes per incremental session over 8 sessions.

For the encoder, we use ResNet-18 by default for all datasets, similar to (Tao et al., 2020; Kang et al., 2023). We train the model with an SGD optimizer, a momentum of 0.9, and a batch size of 64. The learning rate is set to 0.1 for CIFAR-100 and miniImageNet and 0.001 for CUB-200. For all experiments, the model is trained on an Nvidia RTX 2080 GPU.

## S2  Comparison to State-of-the-art

In this section, we present the results for CUB-200 and miniImageNet datasets and their comparison to prior works. As we observe from Table S1, our framework outperforms all existing works on CUB-200 and sets a state-of-the-art of 63.60%. Similarly, for miniImageNet in Table S2, our framework performs better than all prior works, showing a new best accuracy of 59.11%.

## S3  Additional Results

### S3.1  Baseline

In this section we discuss sensitivity studies on some of the key hyper-parameters of the baseline method, including the number of training epochs and learning rate. The sensitivity study on the training epochs is presented in Figure S1. Our findings from this study show that the best results for CIFAR-100, CUB-200 and miniImageNet datasets are obtained for training 400, 80, and 80 epochs, respectively. The study on learning rate in Table S3 shows that the best performances are achieved for a learning rate of 0.1 for CIFAR-100, and miniImageNet datasets, and 0.001 for the CUB-200 dataset.

Table S1: Comparison to prior works across the base and incremental sessions on CUB-200.

| Method | Acc. in each session (%) ↑ | | | | | | | | | | |
|---|---|---|---|---|---|---|---|---|---|---|---|
| | 0 | 1 | 2 | 3 | 4 | 5 | 6 | 7 | 8 | 9 | 10 |
| Baseline | 79.57 | 75.59 | 72.23 | 67.53 | 67.49 | 64.73 | 64.41 | 62.69 | 60.54 | 60.85 | 59.88 |
| iCaRL (Rebuffi et al., 2017) | 68.68 | 52.65 | 48.61 | 44.16 | 36.62 | 29.52 | 27.83 | 26.26 | 24.01 | 23.89 | 21.16 |
| EEIL (Castro et al., 2018) | 68.68 | 53.63 | 47.91 | 44.20 | 36.30 | 27.46 | 25.93 | 24.70 | 23.95 | 24.13 | 22.11 |
| TOPIC (Tao et al., 2020) | 68.68 | 62.49 | 54.81 | 49.99 | 45.25 | 41.40 | 38.35 | 35.36 | 32.22 | 28.31 | 26.28 |
| Rebalancing (Hou et al., 2019) | 68.68 | 57.12 | 44.21 | 28.78 | 26.71 | 25.66 | 24.62 | 21.52 | 20.12 | 20.06 | 19.87 |
| SPPR (Zhu et al., 2021b) | 68.68 | 61.85 | 57.43 | 52.68 | 50.19 | 46.88 | 44.65 | 43.07 | 40.17 | 39.63 | 37.33 |
| MetaFSCIL(Chi et al., 2022) | 75.90 | 72.41 | 68.78 | 64.78 | 62.96 | 59.99 | 58.30 | 56.85 | 54.78 | 53.82 | 52.64 |
| F2M (Shi et al., 2021) | 81.07 | 78.16 | 75.57 | 72.89 | 70.86 | 68.17 | 67.01 | 65.26 | 63.36 | 61.76 | 60.26 |
| CEC (Zhang et al., 2021) | 75.85 | 71.94 | 68.50 | 63.50 | 62.43 | 58.27 | 57.73 | 55.81 | 54.83 | 53.52 | 52.28 |
| FACT (Zhou et al., 2022a) | 75.90 | 73.23 | 70.84 | 66.13 | 65.56 | 62.15 | 61.74 | 59.83 | 58.41 | 57.89 | 56.94 |
| LIMIT (Zhou et al., 2022b) | 75.89 | 73.55 | 71.99 | 68.14 | 67.42 | 63.61 | 62.40 | 61.35 | 59.91 | 58.66 | 57.41 |
| SoftNet (Kang et al., 2023) | 78.07 | 74.58 | 71.37 | 67.54 | 65.37 | 62.60 | 61.07 | 59.37 | 57.53 | 57.21 | 56.75 |
| ALICE (Peng et al., 2022a) | 77.40 | 72.70 | 70.60 | 67.20 | 65.90 | 63.40 | 62.90 | 61.90 | 60.50 | 60.60 | 60.10 |
| SAVC (Song et al., 2023) | 81.85 | 77.92 | 74.95 | 70.21 | 69.96 | 67.02 | 66.16 | 65.30 | 63.84 | 63.15 | 62.50 |
| **Ours** | **82.31** | **78.03** | **75.45** | **70.99** | **71.06** | **67.85** | **67.44** | **66.05** | **64.95** | **64.31** | **63.60** |

Table S2: Comparison to prior works across the base and incremental sessions on miniImageNet.

| Method | Acc. in each session (%) ↑ | | | | | | | | |
|---|---|---|---|---|---|---|---|---|---|
| | 0 | 1 | 2 | 3 | 4 | 5 | 6 | 7 | 8 |
| Baseline | 69.08 | 64.4 | 60.22 | 57.08 | 53.8 | 50.88 | 48.42 | 46.54 | 45.08 |
| iCaRL (Rebuffi et al., 2017) | 71.77 | 61.85 | 58.12 | 54.60 | 51.49 | 48.47 | 45.90 | 44.19 | 42.71 |
| Rebalancing (Hou et al., 2019) | 72.30 | 66.37 | 61.00 | 56.93 | 53.31 | 49.93 | 46.47 | 44.13 | 42.19 |
| TOPIC (Tao et al., 2020) | 61.31 | 50.09 | 45.17 | 41.16 | 37.48 | 35.52 | 32.19 | 29.46 | 24.42 |
| EEIL (Castro et al., 2018) | 61.31 | 46.58 | 44.00 | 37.29 | 33.14 | 27.12 | 24.10 | 21.57 | 19.58 |
| FSLL (Mazumder et al., 2021) | 66.48 | 61.75 | 58.16 | 54.16 | 51.10 | 48.53 | 46.54 | 44.20 | 42.28 |
| FSLL+SS (Mazumder et al., 2021) | 68.85 | 63.14 | 59.24 | 55.23 | 52.24 | 49.65 | 47.74 | 45.23 | 43.92 |
| F2M (Shi et al., 2021) | 72.05 | 67.47 | 63.16 | 59.70 | 56.71 | 53.77 | 51.11 | 49.21 | 47.84 |
| CEC (Zhang et al., 2021) | 72.00 | 66.83 | 62.97 | 59.43 | 56.70 | 53.73 | 51.19 | 49.24 | 47.63 |
| MetaFSCIL (Chi et al., 2022) | 72.04 | 67.94 | 63.77 | 60.29 | 57.58 | 55.16 | 52.90 | 50.79 | 49.19 |
| C-FSCIL (Hersche et al., 2022) | 76.40 | 71.14 | 66.46 | 63.29 | 60.42 | 57.46 | 54.78 | 53.11 | 51.41 |
| FACT (Zhou et al., 2022a) | 72.56 | 69.63 | 66.38 | 62.77 | 60.60 | 57.33 | 54.34 | 52.16 | 50.49 |
| CLOM (Zou et al., 2022) | 73.08 | 68.09 | 64.16 | 60.41 | 57.41 | 54.29 | 51.54 | 49.37 | 48.00 |
| LIMIT (Zhou et al., 2022b) | 72.32 | 68.47 | 64.30 | 60.78 | 57.95 | 55.07 | 52.70 | 50.72 | 49.19 |
| SoftNet (Kang et al., 2023) | 79.77 | 75.08 | 70.59 | 66.93 | 64.00 | 61.00 | 57.81 | 55.81 | 54.68 |
| ALICE (Peng et al., 2022a) | 80.60 | 70.60 | 67.40 | 64.50 | 62.50 | 60.00 | 57.80 | 56.80 | 55.70 |
| SAVC (Song et al., 2023) | 81.12 | 76.14 | 72.43 | 68.92 | 66.48 | 62.95 | 59.92 | 58.39 | 57.11 |
| **Ours** | **84.3** | **79.59** | **75.49** | **71.4** | **68.45** | **65.129** | **62.2** | **60.52** | **59.11** |

### S3.2 Stability Tricks

In this section, we discuss additional results on the stability tricks. In Figure S2, we study the sensitivity towards the number of epochs when training with the SupCon loss. This study shows that optimal performance with the SupCon loss is observed for a relatively larger number of epochs in comparison to the baseline. More specifically, the best results for CIFAR-100, CUB-200, and miniImageNet are observed when trained for 500, 100, and 120 epochs, respectively.

Next, we discuss the experiments on pre-assigning prototypes. As discussed in Section 3.3, we assign the prototype after training for a pre-defined number of epochs. In Table S4, we study the optimal epoch before assigning the prototype, defined as a factor of total epochs. For instance, a value of 0.1 means the prototypes are assigned after training for 10% of the total number of epochs. As we find from this table, assigning a prototype at the beginning of the training does not yield the best performance for any dataset. The best results for the CIFAR-100, CUB-200, and miniImageNet are obtained for the epoch factor of 0.1, 0.5, and 0.5, respectively.

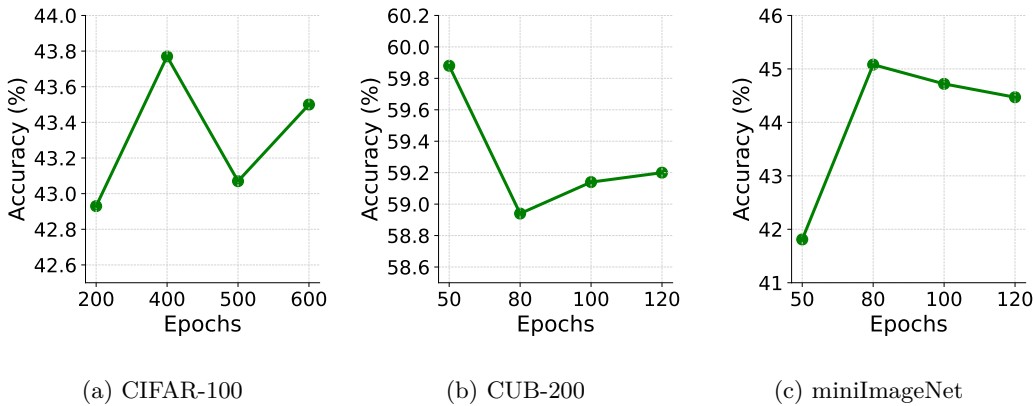

(a) CIFAR-100      (b) CUB-200      (c) miniImageNet

Figure S1: Sensitivity study on the number of training epochs for the baseline method.

Table S3: Sensitivity study on the learning rate for the baseline method.

| LR | CIFAR-100 | CUB-200 | miniIN |
|---|---|---|---|
| 0.5 | 29.36 | 55.02 | 34.74 |
| 0.1 | **43.77** | 57.75 | **45.08** |
| 0.01 | 42.73 | 58.33 | 43.42 |
| 0.001 | 39.47 | **59.88** | 32.54 |
| 0.0001 | 19.63 | 50.17 | 16.01 |

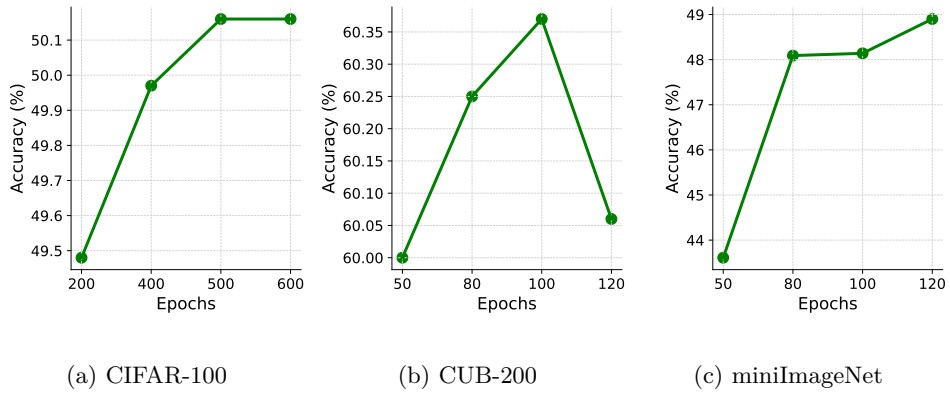

(a) CIFAR-100      (b) CUB-200      (c) miniImageNet

Figure S2: Sensitivity study on the number of training epochs for SupCon.

### S3.3 Adaptability Tricks

In this section, we discuss the experiments on the adaptability tricks. As discussed in Section 3.5, we only tune a few layers of the encoder during the incremental fine-tuning, keeping the remaining layers frozen. Table S5 presents the results for fine-tuning different portions of the pre-trained encoder. As seen in this table, the best results are observed when only the last ResNet block is turned, while the worst results are consistently observed for tuning the full encoder.

### S3.4 Training Tricks

Finally, we discuss the experiments on the training tricks. In Table S6, we show the results for training different encoders on all the datasets. As we observe in this table, increasing the model size in our framework

Table S4: Sensitivity study on the epoch factor for prototypes.

| Epoch factor | CIFAR-100 | CUB-200 | MiniIN |
|:---:|:---:|:---:|:---:|
| 0 | 49.66 | 60.50 | 51.01 |
| 0.1 | 49.45 | **60.74** | **51.10** |
| 0.5 | **49.73** | 60.24 | 50.87 |
| 0.75 | 49.35 | 60.11 | 50.75 |

Table S5: Sensitivity study on tuning different portions of the encoder.

| Tunable parameters | CIFAR-100 | CUB-200 | MiniIN |
|:---:|:---:|:---:|:---:|
| Full encoder | 49.11 | 59.16 | 51.19 |
| Last ResNet block | **52.51** | **62.27** | **55.82** |
| Last 2 ResNet blocks | 52.02 | 61.50 | 55.18 |

Table S6: Study on different encoders.

| Encoder | CIFAR-100 | CUB-200 | MiniIN |
|:---:|:---:|:---:|:---:|
| ResNet-18 | 58.12 | 63.10 | 57.85 |
| ResNet-34 | 58.44 | 63.17 | 57.90 |
| ResNet-50 | **58.52** | **63.55** | **57.99** |
| ResNet-101 | 57.95 | 63.19 | 57.55 |

Table S7: Comparison to prior works with different shots of data.

| Method | Acc. in each session (%) ↑ | | | | | | | | |
|---|---|---|---|---|---|---|---|---|---|
| | **0** | **1** | **2** | **3** | **4** | **5** | **6** | **7** | **8** |
| **1-shots** | | | | | | | | | |
| SAVC (Song et al., 2023) | 79.85 | 71.29 | 64.11 | 57.08 | 53.45 | 50.45 | 48.74 | 46.54 | 44.45 |
| SoftNet (Kang et al., 2023) | 79.88 | 72.01 | 65.01 | 59.01 | 55.01 | 52.01 | 50.01 | 48.01 | 46.01 |
| **Ours** | **80.25** | **76.01** | **71.87** | **67.24** | **63.56** | **59.82** | **56.62** | **54.11** | **52.84** |
| **2-shots** | | | | | | | | | |
| SAVC (Song et al., 2023) | 79.85 | 72.00 | 65.45 | 59.17 | 55.25 | 52.2 | 50.16 | 48.11 | 46.94 |
| SoftNet (Kang et al., 2023) | 79.88 | 73.02 | 66.01 | 60.01 | 56.01 | 53.01 | 51.01 | 49.01 | 47.01 |
| **Ours** | **80.25** | **77.63** | **73.071** | **68.53** | **64.51** | **60.67** | **58.41** | **55.98** | **54.15** |
| **5-shots** | | | | | | | | | |
| SAVC (Song et al., 2023) | 79.85 | 73.70 | 69.37 | 65.28 | 61.91 | 59.27 | 57.24 | 54.97 | 53.12 |
| SoftNet (Kang et al., 2023) | 79.88 | 75.54 | 71.64 | 67.47 | 64.45 | 61.09 | 59.07 | 57.29 | 55.33 |
| **Ours** | **80.25** | **77.20** | **75.09** | **70.82** | **67.83** | **64.86** | **62.73** | **60.52** | **58.75** |
| **10-shots** | | | | | | | | | |
| SAVC (Song et al., 2023) | 79.85 | 75.45 | 72.18 | 68.26 | 65.66 | 62.15 | 60.27 | 58.19 | 56.01 |
| SoftNet (Kang et al., 2023) | 79.88 | 77.28 | 73.15 | 69.34 | 66.14 | 63.38 | 61.27 | 59.33 | 57.14 |
| **Ours** | **80.25** | **78.10** | **76.39** | **72.02** | **69.03** | **65.96** | **64.00** | **61.92** | **60.25** |

improves the performance up to a certain model size. Notably, we observe that the best performance for all the datasets is obtained with ResNet-50.

### S3.5 Performance on Different Shots

In this section, we present additional results on different shots. More specifically, we present the results for 1-, 2-, 5-, and 10-shots on the CIFAR-100 dataset. The results of this study are presented in Table S7. As we see from this table, our framework outperforms prior works in data-scarce settings. Specifically, in a 1-shot setting, our framework outperforms the previous state-of-the-art by a significant margin of 6.83%. In the 2-shot setting, the difference increases slightly to 7.14%. In a 5-shot setting, the improvement over existing methods decreases to 3.42%. Finally, we find a boost in performance for all the methods when we increase the labelled samples to a 10-shot setting, with our framework showing 3.11% improvement over the SoftNet.

