# OpenReview forum: "A Bag of Tricks for Few-Shot Class-Incremental Learning"
_TMLR — Accepted by TMLR_

### Review · Reviewer_rHCM · 2024-06-25

**Summary Of Contributions:**

The paper proposes to combine 8 different tricks to improve the performance of few-shot class-incremental learning (FSCIL). The tricks are grouped into 3 groups: stability, adaptability, and training techniques. The paper then performs experiments and some in-depth analysis on 3 benchmark datasets to evaluate the effectiveness of these tricks on FSCIL, showing they can achieve a new state-of-the-art accuracy for this problem.

**Audience:**

Yes

**Broader Impact Concerns:**

No concerns on broader impacts and ethical implications.

**Claims And Evidence:**

Yes

**Requested Changes:**

I would suggest a major change in the presentation of the proposed approach, especially by consolidating the tricks and showing a full picture of how the tricks would be combined together. More specifically (*all these points are critical*):
- The authors could provide a figure showing the order where the tricks are performed if all of them are used together. In this figure, we could use different colors for different categories of tricks.
- I suggest to remove the larger encoder trick since it seems not a significant step.
- For the adaptability tricks, I suggest to combine both incremental fine-tuning and SubSet tuning into a single trick (with some new name), since these two tricks are best used together, as shown in the experiment results (e.g., in figure 3).
- The paper should add more details on the ETF and pseudo-classes tricks, especially their training losses and how these losses are combined together. These information could also be added into the overall figure above.
- Some experiment results are not consistent. For example, in table 3, the accuracy without training tricks is 57.85, but this accuracy is 58.12 in table 1. Besides, further removing the adaptability tricks results in 54.79 in table 3 but 51.65 in table 1. There should be a throughout check to ensure consistency of the results and if possible, also including the error ranges of the results.

Besides these major changes, there are other minor changes that need to be addressed. Specifically:
- In Section 3.1, since $T$ is a number, each session $t$ should be in $\\{ 1, 2, \ldots, T \\}$.
- On page 4, the paragraph after Eq (1), $y_i = y_j$ should be in the subscript of the indicator function.
- In Eq (2), the operator $\odot$ should be defined.
- In Eq (3) and the paragraph after that, the indicator function should be used consistently with that in Eq (1); for example, without (or with) the square brackets in the subscript. The temperature $\tau$ should be introduced in Eq (1) as well.
- In the legends of Figure 2, it should be "ETF" instead of "EFT".
- In the paragraph right after the caption of figure 2, there should be a closing parenthesis after "illustrated in Figure 2b".
- In Section 4.3, some references to Section 4.2 should be "Section 4.2" instead of "Appendix 4.2".
- On page 8, the paragraph before the Adaptability Tricks paragraph, the numbers 51.21%, 62.27%, 55.82% are not consistent with Table 1.
- In Figure 2, why do you choose to report cumulative probability instead of a simple average distance? Some explanations are needed here.

**Strengths And Weaknesses:**

**Strengths:**
- Although each of the individual tricks considered in this paper is not novel, the idea of combining them into a single framework to improve the performance of FSCIL is sensible and has not been considered before.
- The experiment results show the proposed approach has better accuracy than previous state-of-the-art methods such as SAVC and SoftNet.
- The paper provides some in-depth analysis of the inter-class distance, intra-class distance, and class separation when adding the stability tricks on the CIFAR-100 dataset. The paper also visualizes the effects of the adaptability tricks on the class separation on this dataset. These analyses are helpful for understanding the proposed approach.

**Weaknesses:**
- One main weakness of the paper is the unclear presentation of the proposed approach. In the current manuscript, each of the tricks is just presented separately, and it is not clear how all these tricks can be combined into a single framework. For instance, in their original papers, the SupCon, ETF, and pseudo-classes all have different training losses. It is not clear how these losses can be combined into a single framework when these tricks are used together.
- Related to the previous point, in the training tricks, both contrastive pre-training and additional learning signal are self-supervised learning steps performed before the base training session. However, it is not clear to me how these tricks can be used together (i.e., what would the training loss be when the two self-supervised learning methods are used together?).
- Additionally, it is also not clear how the fine-tuning and SubNet tuning tricks can be combined, since they are all fine-tuning different parts of the network.
- I am not very sure that the change from ResNet-18 to ResNet-50 could be qualified as a major trick. This change seems trivial and the accuracy improvements in the experiments are also not great.

---

> ### Author Response · Authors · 2024-07-24
>
> We thank the reviewer for their valuable feedback and comments. Below we address each point and highlight the related changes in the revised manuscript with the color **blue**.
>
> > Order in which the tricks are performed
>
> Thanks for the suggestion. The order in which the tricks are performed were previously shown in Table 1. We have now revised this to better demonstrate the **order** in which the tricks are used (by indenting each trick following the previous trick), along with their cumulative impacts. We also added the *Stage* of the training to the paper. The corresponding text on page 8 of the paper has also been revised accordingly. We could not find a way to effectively show this in a figure format (which is what you asked for) instead of the table - we hope the reviewer finds this acceptable.
>
>
> > Remove the larger encoder trick
>
> As per the suggestion, we have now removed the larger encoder from the list of training tricks. However, **Reviewer YMFp** has suggested including the performance on newer and larger encoders, namely ViT. To this end, we have reported the results on both large (ViT, ResNet-50) and small encoders (ResNet-18, ResNet-20) in Table 3 to show that the tricks generalize to larger encoders. However, kindly note that we do not present these under the category of any specific tricks, as you suggested.
>
> > Combine both incremental fine-tuning and SubSet tuning into a single trick
>
> As per your suggestion, we have now combined the two adaptability tricks under the new name of `Incremental SubNet Tuning.' Under the combined trick, the original first adaptability trick (Incremental Fine-tuning) dictates the fine-tuning of the classifier with a lower LR, while the original second adaptability trick (SubNet Tuning) tunes a subset of the encoder. This is now presented on Page 5 of the revised manuscript and also reflected throughout the results section of the paper.
>
> > More details on the ETF and pseudo-classes tricks
>
> In the revised manuscript, We have expanded the description of the ETF and pseudo-class tricks (see Page 5). In summary, the Pre-assigning Prototypes optimize an MSE loss to align the learned prototype with the pre-assigned prototype. Including pseudo-classes tricks simply increases the number of classes by considering the special transformation of a class as a different class and does not include any additional loss function to optimize.
>
> > Consistency in the experiment results
>
> Thank you for pointing out this oversight. We mistakenly reported the accuracy of miniImageNet instead of CIFAR-100 for the mentioned two settings. We have updated Table 2 accordingly. Please note that some of the results have been further changed due to removing the large encoder trick.
>
>
> > Cumulative probability vs. simple average distance
>
> We plot the cumulative probability of inter-class and intra-class distance instead of the average since the average distance can be misleading due to the presence of outliers, whereas the cumulative probability provides a more robust and nuanced representation of the distribution of distances. This follows prior works such as Song el al. CVPR 2023, which also reports a similar plot.
>
>
> > Minor Changes
>
> Thank you for suggesting the minor changes. We have revised the manuscript and addressed all of these points.

---

### Review · Reviewer_R2Ls · 2024-06-29

**Summary Of Contributions:**

This paper presents a comprehensive framework aimed at enhancing both stability and adaptability in Few-Shot Class-Incremental Learning (FSCIL). The framework is categorized into three main strategies: stability tricks, adaptability tricks, and training tricks. Stability tricks are designed to minimize forgetting by improving class separation within the embedding space. Adaptability tricks facilitate the learning of new classes through targeted fine-tuning methods. Training tricks aim to boost overall model performance while maintaining stability and adaptability. The framework’s effectiveness is validated through extensive testing on CIFAR-100, CUB-200, and miniImageNet, where it demonstrates superior performance over existing methods and sets a robust baseline for future FSCIL research.

**Audience:**

Yes

**Claims And Evidence:**

Yes

**Requested Changes:**

1.	The authors should delineate the innovative aspects of the tricks introduced in this paper, specifically highlighting how these adaptations or enhancements differ from or improve upon previous implementations.

2.	In the conclusion, the paper discusses a limitation related to integrating multiple tricks within a single framework. It would be beneficial for the authors to elaborate on the challenges faced in this integration and how they impact the framework’s adaptability or scalability.

3.	Figure 1 presupposes that incremental classes differ from base classes. The authors should explore and discuss the effectiveness of stability tricks when incremental classes closely resemble or are identical to base classes. This investigation would greatly enhance understanding of the framework’s robustness and versatility.

**Strengths And Weaknesses:**

[Strengths]

The paper is well-supported by numerous detailed figures that effectively illustrate the main concepts and enhance understanding. Additionally, the authors provide a thorough comparison with other existing methods, highlighting the framework’s advancements in the field.

[Weaknesses]

1.	The paper lacks clarity regarding the novelty of the proposed tricks. Many of the techniques discussed appear to have been previously utilized in other studies, making it difficult to ascertain the unique contributions of this work.
2.	The research is exclusively conducted on image classification tasks using specific datasets. This specialization limits understanding of the framework’s applicability to other domains, such as text or audio processing.

---

> ### Author Response · Authors · 2024-07-24
>
> We thank the reviewer for their valuable feedback and comments. Below we address each point and highlight the related changes in the revised manuscript with the colour **green**.
>
> > Delineate the innovative aspects of the tricks
>
> The key innovation of our framework is combining a set of modules (tricks), which improve stability, adaptability, and overall performance at the same time. As discussed in the related work, the stability-adaptability trade-off is a well-known phenomenon in the FSCIL literature, where improvement in one aspect often comes with the cost of another, and it is difficult to improve both aspects together. While we did not propose any new components, no prior work has explored such modules (tricks) together in the context of FSCIL under a unified framework. This has now been expressed in the paper on Pages 1 and 2.
>
> > Limitation related to integrating tricks within a single framework
>
> We would like to clarify that our strength is indeed in bringing together all the tricks discussed in our paper, together under one roof. To present the conclusion section more clearly, we have now merged the limitations section inside the conclusion section and renamed it as 'Conclusion and Future Work'. Please refer to page 13 of the revised manuscript.
>
>
> > Effectiveness of stability tricks when incremental classes closely resemble the base classes
>
> Thank you for this interesting question. According to the setup adopted in the literature for FSCIL (Song et al. CVPR 2023, Kang et al. ICLR 2023), each of the new classes in the incremental session is different from the classes seen so far. Therefore, incremental classes are generally not exactly identical to the base classes. However, some of the incremental classes can be *similar* to the base classes as you suggested. For instance, in CIFAR-100, 'Orchid' is a class of flower that is part of the base class, while 'Sun-flower' is another flower that is part of the incremental class. A few similar pairs of examples of base classes with similarities to incremental classes are 'Lion'-'Tigar', 'Bus'-'Streetcar', and 'Shark'-'Dolphin', respectively. As per your comment, to understand the effectiveness of the stability on *closely related classes*, we report the average accuracy on the above-mentioned base and incremental classes with and without our tricks below:
>
> | Method | Base          | Incremental |
> | --------|--------------|------------------|
> | Baseline| 65.3\%       | 19.7\%           |
> | Ours    | 74.5\%       | 30.4\%           |
>
>
> As we find from this experiment, even for semantically similar incremental classes, our proposed framework is capable of improving the performance of both the base classes (improved stability) and the incremental ones (improved adaptability). This table and discussion have now been added to Page 12 of the revised paper.

---

### Review · Reviewer_YMFp · 2024-07-13

**Summary Of Contributions:**

The paper introduces a unified framework that integrates eight highly influential techniques, categorized as stability, adaptability, and training tricks, to enhance both the stability (reducing forgetting of previously learned classes) and adaptability (effectively learning new classes) of FSCIL models. The authors conduct a thorough analysis of each trick's contribution towards stability and adaptability in the context of FSCIL, providing valuable insights into their individual and combined effects. This analysis sheds light on the complex interplay between these techniques and their impact on overall performance. They experiment with three benchmarks (CIFAR-100, CUB-200, and miniImageNet), the proposed framework demonstrably outperforms existing state-of-the-art methods, however I have concerns regarding the experimental setup used in the paper. The authors also study a low-shot settings (1-shot and 2-shot) to assess the framework's effectiveness under extreme data scarcity.

**Audience:**

Yes

**Broader Impact Concerns:**

N\A

**Claims And Evidence:**

Yes

**Requested Changes:**

1. Add some experiments related to some newer models like ViT and compare with methods like "Learning to Prompt for Continual Learning" and followups.
2. Provide more detailed explanation and justification for the specific SubNet extraction method used. Do they first optimize to find the subnetwork? or find the subnetwork in some other way and then just train it?
3. Further investigation into adaptability: Given the persistent gap between performance on base and novel classes, the authors could further analyze the limitations of current adaptability tricks and explore potential avenues for bridging this gap.
4. See Weakness section.

**Strengths And Weaknesses:**

Strengths:
1. The paper systematically explores a wide range of techniques for improving FSCIL performance.
2. The authors provide detailed insights into the individual contributions of each trick and their impact on stability and adaptability.
3. The proposed framework achieves significant performance improvements over previous methods.
4. Exploring 1-shot and 2-shot scenarios demonstrates the framework's robustness under data scarcity.

Weakness:
1. I think the biggest weakness of the paper is that it uses models and benchmarks which are extremely outdated and almost toyish. Since pretrained models have come into the picture the field of continual learning has very dramatically and drastically changed. The few shot adaptation in most cases is trivial by just prompting the model. So in the light of the progress in the field since 2017 I find that the insight from the paper might not be applicable in the settings in which people mostly find them these days. For example, from my experience the insights from resnet model and MNIST/cifar tasks would not tranfer at all when we have models like ViT or even T5 kind of models. The paper does a really good job at the through analysis however my worry is that none of these insights would be useful.
2. The paper primarily combines existing techniques. While the unified framework and analysis are valuable, the individual techniques are not novel.
3. The explanation of the SubNet tuning technique could be further elaborated for better clarity.
4. The paper acknowledges the higher computational cost but provides limited analysis or discussion on its implications. I see that in Table 6 they provide some FPS throughput however It would be great if they can provide wall clock time for the experiments where there are additional components involved.

---

> ### Author Response · Authors · 2024-07-24
>
> We thank the reviewer for their valuable feedback and comments. Below, we address each point and highlight the related changes in the revised manuscript with the color **purple**.
>
> > Models and benchmarks which are outdated and almost toyish...Add experiments related to some newer models like ViT.
>
> We would like to point out that we followed the same problem setup and encoders used by all the prior works on the Few-shot Class-incremental Learning (FSCIL), including papers that have been published in recent venues like ICLR 2023 and CVPR 2023. Our focus in this paper is to improve the stability, adaptability, and overall performance of FSCIL while following the same benchmarks and problem setups, as well as encoder backbones to ensure a fair comparison. Please note that the tricks in our framework are independent of the encoder's choice and can be easily adapted to any encoder, including ViT. To address your concern regarding ViTs, in the Table below (Table 3 of the revised manuscript) of the revised manuscript, we have included the results of our framework on larger encoders, including ResNet-50, ViT-B/16, and ViT-B/16 pre-trained on ImageNet. The results from this study show that the proposed tricks not only transfer to larger encoders and ViTs, but the performance gains are even larger than those of smaller networks like ResNet-20.
> For instance, ResNet-50 shows a 20.74\% improvement compared to 14.78\% in ResNet-20. Similarly, ViT-B/16 with pre-trained and randomly initialized encoders shows 19.58\% and 15.75\% improvements over the baseline. This is due to the fact that without the stability and adaptability tricks from our framework, a larger encoder with more parameters is more prone to overfitting, making it difficult to learn incremental classes in a few-shot learning setting. With our proposed framework, the larger encoders show a large improvement in performance. This discussion is added to Page 10 of the revised manuscript.
>
> Encoder   | Baseline| Ours   |
> ----------|---------|--------|
> ResNet-20    | 43.77 | 58.55 |
> ResNet-50    | 38.01 | 58.75 |
> ViT-B/16     | 32.14 | 47.89 |
> ViT-B/16 (pre-trained)| 50.14 | 69.72 |
>
>
>
> > Compare with methods like "Learning to Prompt for Continual Learning" and follow-ups.
>
> Please note that the mentioned paper (Learning to Prompt for Continual Learning) is a method for fine-tuning a foundation model for continual learning in a fully-supervised setting, while the focus of our work is on few-shot class-incremental learning (FSCIL), which is a different benchmark than fully-supervised continual learning. As a result, we can’t make a direct comparison to this method. Additionally, fine-tuning a pre-trained model (full fine-tuning or parameter-efficient fine-tuning) requires different training protocols, hyper-parameters, and consideration of domain discrepancy between the source (pre-trained data) and target domain, exploring which are beyond the problem definition or the focus of our current work. We carefully studied prior works in this area and concluded that all SOTA works make clear distinctions between these two areas and do not make such comparisons due to the considerable difference between few-shot learning and fully supervised learning. However, to address your concern regarding **larger** backbones and **pre-trained** encoders, we have performed additional experiments and revised the paper accordingly (please see the previous comment).
>
>
> > While the unified framework and analysis are valuable, the individual techniques are not novel
>
> The main contribution of our work lies in presenting a framework that combines a set of known tricks that jointly improve both stability and adaptability, which, as discussed in the literature review, is a difficult task since improved stability or adaptability often hampers the other aspect. Additionally, We provide extensive analysis of these tricks for a better understanding of their impact beyond the final accuracy, including the impact on stability and adaptability, class separation in the learned embedding space, and performance improvement on base and novel classes. Furthermore, we present a detailed study and comparison to prior works on different aspects that are not explored in the existing literature, including low-shot (1-, 2-shot) performance, results on a dataset with a large number of classes (InageNet-1k), performance on larger encoders (e.g. ResNet-50, ViT-B/16), performance on fine-tuning a pre-trained foundation model (ViT-B/16) [thank you for this suggestion], performance with semantically similar incremental classes (included in revised manuscript). We believe the unified framework and the extensive set of experiments add value to further development of this area.
> We hope that this discussion clarifies the contribution of this paper to the challenging field of FSCIL. We add this discussion to the conclusion of the revised manuscript on Page 12.

---

> > ### Author Response · Authors · 2024-07-24
> >
> > > Additional details of SubNet Tuning
> >
> > Using the lottery ticket hypothesis, the SubNet Tuning trick finds a sub-network on the trained encoder after the base session. According to the lottery ticket hypothesis, there exists a subnetwork of a dense network that is as permanent as the whole network. Finding a subnet after the base session and tuning only the remaining parameters ensures the performance of the base session is preserved while providing improved adaptability for learning the new classes. We followed Kang et al. 2023 to extract this SubNet using the forums described in Eq 3 of the manuscript. Please find this discussion on Pages 5 and 6. Please note that, as per the suggestion from Reviewer rHCM we have combined the two adaptability tricks into one under the new name 'Incremental SubNet Tuning'.
> >
> >
> > > Regarding higher computational cost
> >
> > We originally mentioned higher computational complexity as a limitation of our work, which was caused by our **larger encoder** trick (compared to prior works). As per the suggestion from Reviewer rHCM, we have now removed that trick from our framework, so our current model is as efficient as the previous SOTA during inference (please see Table 6 in the paper). As per your suggestion, we have now also added the wall clock time to Table 6 on Page 12.
> >
> >
> > > Further investigation into adaptability
> >
> > We would like to point out the fact that our framework has already improved the performance of incremental classes by 20.2\% (10.8 to 31.0) and reduced the gap between the base and incremental classes from 55.0\% to 45.2\% while not hampering the performance of the base classes. The remaining gap between the performance could be caused by the fact that the base session has more samples, leading to better learning of the base classes compared to incremental classes. We acknowledge that this is indeed an important area for inquiry and future research, and we hope that our work can further facilitate improvements in this area. We add this discussion to the conclusion on Page 13.

---

### Author Response · Authors · 2024-07-24

We sincerely thank the reviewers for their time and for providing constructive feedback. We are happy to see the overall engaging comments given by all the reviewers. We are glad to know that reviewers find the paper valuable as the idea is sensible and has not been considered before (Reviewer rHCM), systematically explores a wide range of techniques (Reviewer YMFp), provides insights into each trick on stability and adaptability (Reviewer YMFp, rHCM), achieves significant performance improvements (Reviewers YMFp, R2Ls, rHCM), shows robustness under data scarcity such as 1-shot (Reviewer YMFp), and well-supported by numerous detailed figures that effectively illustrate the main concepts (Reviewer R2Ls). We have carefully addressed all the concerns raised by the reviewers under the individual response section. Following, we provide a *summary* of our responses.

**—Combining and removing some tricks:** As per the suggestion of **Reviewer rHCM**, we have combined two 'adaptability' tricks into one trick and removed a 'larger encoder' trick from the list of training tricks.

**—Additional analysis:** We have included a new analysis of the proposed framework's performance for incremental classes with semantic similarity to base classes.

**—Generalizability to larger and pre-trained encoders:** We have included additional experiments to show the generalizability of our framework to large transformer-based encoders and pre-trained foundation models.

---

### Decision · Action_Editor_fc5A · 2024-08-22

**Recommendation:** Accept with minor revision

**Comment:**

This paper presents a unified framework that integrates multiple techniques, termed "tricks," to improve Few-Shot Class-Incremental Learning (FSCIL). The proposed framework categorizes these techniques into stability tricks, adaptability tricks, and training tricks, each contributing to the overall enhancement of model performance, stability, and adaptability. The authors conduct extensive experiments on standard benchmarks, demonstrating significant performance improvements and providing detailed analyses of the effects of these tricks on the model's learning process.

The reviewers expressed concerns regarding the experimental setup, particularly the use of outdated models and benchmarks. In response, the authors have added more experiments, including those involving larger and more modern encoders like ViT. If possible, it would be beneficial for the authors to further strengthen the paper by including additional experiments that leverage pre-trained large models.

**Audience:**

Yes

**Claims And Evidence:**

Yes